



# Biogenic isoprenoid emissions under drought stress: Different responses for isoprene and terpenes

Boris Bonn[1], Ruth-Kristina Magh[1], Joseph Rombach[1], Jürgen Kreuzwieser[1]

[1]Chair of Ecosystem Physiology/Chair of Tree Physiology, Faculty of Environment and Natural Resources, Albert-Ludwigs-Universität, Georges-Köhler-Allee 053, D-79110 Freiburg i.Br., Germany

*Correspondence to*: Boris Bonn (boris.bonn@ctp.uni-freiburg.de)

**Abstract.** Emissions of volatile organic compounds (VOCs) by biogenic sources depend on different environmental conditions. Besides temperature and photosynthetic active radiation (PAR), the available soil water can be a major factor, controlling the emission flux. This factor is expected to become more important under future climate conditions including prolonged drying-wetting cycles. In this paper we use results of available studies on different tree types to set up a parameterization describing the influence of soil water availability (SWA) on different isoprenoid emission rates. Investigating SWA effects on isoprene ($C_5H_8$), mono- ($C_{10}H_{16}$) and sesquiterpene ($C_{15}H_{24}$) emissions separately, it is obvious that different plant processes seem to control the individual emission fluxes providing a measure of plants to react on stresses and to interact. The SWA impact on isoprene emissions is well described by a biological growth type curve, while the sum of monoterpenes displays a hydraulic conductivity pattern reflecting the plants stomata opening. However, emissions of individual monoterpene structures behave differently to the total sum, i.e. the emissions of some increase whereas of others decline at decreasing SWA. In addition to a rather similar behaviour as of monoterpene emissions, total sesquiterpene fluxes of species adapted to drought stress tend to reveal a rise close to the wilting point protecting against oxidative damages. Considering further VOCs too, the total sum of VOCs tends to increase at the start of severe drought conditions until resources decline. On the contrary, OH and ozone reactivity enhance. Based on these observations a set of plant protection mechanism displays for drought stress and implies notable feedbacks on atmospheric processes such as ozone, aerosol particles and cloud properties. With progressing length of drought periods declining storage pools and plant structure effects yield different emission mixtures and strengths. This drought feedback effect is definitely worth consideration in climate feedback descriptions and for accurate climate predictions.

## 1 Introduction

The exchange of biogenic VOCs ($E$) is known to represent the largest contribution to global carbon flux besides carbon dioxide ($CO_2$) and methane ($CH_4$)(IPCC, 2014; Niinemets et al., 2014; Holopainen et al., 2017). The total amount of individual BVOC fluxes, i.e. emission and corresponding deposition rates (exchange = emission+deposition), is linked to their production (*de novo* synthesis or online) and storage (offline) capacities of individual plant types and species (Ghirardo et al., 2010) and is



additionally affected by abiotic and biotic conditions. These include the temperatures of vegetation ($T_{veg}$) and soil ($T_{soil}$), photosynthetic active radiation ($PAR$), ambient $CO_2$ mixing ratio, defense against herbivores (Manninen et al., 1998) and reactive air pollutants (Bourtsoukidis et al., 2012, plant-plant communication and competition, fire and drought (Lappalainen et al., 2009; Penuelas et al., 2010; Guenther et al., 2012). While BVOCs represent a large variability of different structures and species (Goldstein and Galbally, 2007; Laothawornkitkul et al., 2009), it is commonly accepted that isoprene ($C_5H_8$) and

monoterpenes ($C_{10}H_{16}$) display the majority (38-50% and 30%, respectively) of total BVOC exchange (1-1.3 Pg yr$^{-1}$) (Goldstein and Galbally, 2007; Guenther et al., 1995; Guenther et al., 2012), if methane is excluded. The exchange $E$ is commonly described by the exchange at standard conditions $E_0$ ($T = 30 \ °C$) and scaled by factors for individual driving forces such as temperature ($T$), light ($L$) (Guenther et al., 1995; 2006;2012) or soil water content (SWC), here denoted as $SM$ in the index in accordance with the formulation of Guenther et al. (2006; 2012):


$$E = E_0 \cdot \gamma_T \cdot \gamma_L \cdot \gamma_{CO2} \cdot \gamma_{SM} \qquad\qquad\qquad (1)$$

Please find an overview of all the abbreviated terms and parameters in Table 1. Some of the scaling factors of driving forces ($\gamma$) for exchange are reasonably well-described regarding temperature and PAR, while other parameters like soil water

availability ($SWA$) are mostly ignored due to a lack of understanding of influencing plant processes, although a simplified parameterisation exists (Guenther et al., 2012). This neglect is confronted with changing climate conditions such as warming of 0.15 K/20 years (Schoelzel et al., 2011) with predicted extending drought periods in Central European forest ecosystems such as the Black Forest in Southern Germany (Keuler et al., 2016; Kreienkamp et al., 2018). So far a single simplified formulation using a linear increase from no emission at the permanent wilting point ($PWP$) to maximum emission shortly

above this point is commonly applied for different conditions. This is based on an experiment of poplar (Pegoraro et al., 2004a), analyzing isoprene only. The changing soil water conditions and the resulting change in BVOC exchange fluxes are expected to influence plant responses and protection capacities (Penuelas et al., 2010; Rennenberg et al., 2006), the exchange following up ozone formation strength and further climate feedback processes (Bonn et al., 2014). In this study we aim at investigating the individual behaviour of isoprene, mono- (MT) and sesquiterpene (SQT) exchange fluxes and their correlation with $SWA$

for available tree species to allow identification of processes controlling the emission. As European beech (*Fagus sylvatica*), which is $SWA$ sensitive (Gessler et al., 2007; Dalsgaard et al., 2011), is one of the most common tree species in Central Europe and of economic importance (Kändler et al., 2016), we will focus on the impact of the $SWA$ effect on emissions.





## 2 Materials and methods

### 2.1 Review of available studies and transfer between different water content parameters used

In order to develop an advanced description of limited soil water on BVOC exchange, available studies were collected, focussing on isoprenoid emissions in the context of different drought conditions. These studies include different tree types and several herbs predominantly under controlled conditions and for selected compounds or compound groups (Table 2).

### 2.2 Adapting available studies to a comparative scheme

#### 2.2.1 Different conditions of individual studies

First, available studies on BVOC exchange for different tree species at different soil water conditions have been collected. It is important to note that these studies have been described in various ways using

(a)  different soil water parameters ($SWA$, soil water content ($SWC$) and water matrix potential ($\psi_m$)),
(b)  different plant ages,
(c)  different soil types,
(d)  different lengths of investigation and time resolutions,
(e)  with and without re-watering and
(f)  different environments (laboratory, greenhouse and ambient).

The named different parameters complicate a direct comparison and include the assumption of an applicable transfer of
laboratory or greenhouse experimental results with predominantly seedlings to ambient conditions, a transfer including notable challenges and questions as discussed e.g. by Niinemets (2010).

#### 2.2.2 Different parameters used for describing the effect of soil water

The plants access to soil water is best described by the accessible water $SWA$ (index '%': fraction of accessible water, index
'v': volumetric amount) or by the suction pressure $\psi_m$. Both parameters denote the soil water status and the ability of a plant to extract water from it (Blume et al., 2010). However, the easiest parameter to quantify soil water conditions by measurement continuously is volumetric $SWC$ ($SWC_v$). Therefore, $SWC_v$ has so far been used to estimate soil water effects on plants processes e.g. by Guenther et al. (2006, 2012). However, different soil types possess different $PWP$ preventing water to be extracted by the plant below. Without the knowledge of the $PWP$ of the soil investigated the information of $SWC_v$ to describe the plant
water access remains incomplete and makes different studies difficult to compare. Therefore, we focus on $SWA_\%$ as primary parameter for describing the soil water effect and provide a set of equations to transfer between the different quantities. This makes different amounts of soil water comparable for different soil conditions and furthermore allows an easy usage in model studies that formerly used $SWC_v$ (Guenther et al., 2006; 2012). To support this all parameterizations are stated by both, i.e. (a) by $SWA_\%$ and (b) by $SWC_v$. The $SWA$ describes the amount of soil water above $PWP$ ($\psi = -4.2$ MPa) up to field capacity





(amount of water held by the soil against gravity, *FK*, $\psi$ = -0.0063 MPa), and is called available or net field capacity (*nFK*)

for the corresponding soil type, and which depends on the soil capacity to fix water (van Genuchten, 1980). During short terms

*SWA* can exceeded *nFK* (e.g. after an intense rain fall), but this excess water infiltrates the soil and is not available to the plant

thereafter. A list of the different parameters used in the available literature is given and the transfer in between is assumed as

(Blume et al., 2010):

$SWA_\% = SWA_v/nFK = (SWC_v\text{-}PWP)/nFK$ (2a)

$SWA_v = SWC_v - PWP$ (2b)

$SWC_v = SWA_v + PWP = SWA_\% \cdot nFK + PWP$ (2c)

$nFK = \max(SWA_v) = FK\text{-}PWP$ (2d)

$FASW_\% = SWC_\% = SWC_v/(nFK + PWP)$ (2e)

$FASW_\%$ and $SWC_\%$ abbreviate the volumetric soil water content relative to its maximum value (field capacity *FK*). Converting

water or matrix potential $\psi$ values is a function of the actual soil mixture of clay, sand and silt, i.e. the different corn size

distributions/classes and the intercorn spaces available for water storage, which can be described by a water retention curve

(van Genuchten diagram (van Genuchten, 1980), see e.g. Blume et al. (2010)):

$SWA_\% = ((SWC_{v,r} + (SWC_{v,max}\text{-}SWC_{v,r})/(1+(a\cdot|\psi\,[MPa]|\cdot 10^4)^n)^{1\text{-}1/n})\text{-}PWP)/nFK$ (3a)

$SWC_v = SWC_{v,r} + (SWC_{v,max}\text{-}SWC_{v,r})/(1+(a\cdot|\psi\,[MPa]|\cdot 10^4)^n)^{1\text{-}1/n}$ (3b)

$SWC_{v,r}$ is the residual *SWC* at completely air dried conditions, $SWC_{v,max}$ is the $SWC_v$ at saturation, a the inverse of the air entry

suction and *n* represents a measure of the pore-size distribution. Representative data for different soil textures can be found

e.g. in Leij et al. (1996) and at http://soilphysics.okstate.edu/software/water/ conductivity.html (access: 2019-01-17).

### 2.2.3 Different soil types, corresponding *PWPs* and *FKs*

*PWP* and *FK* values for different soil types were calculated from the articles information based on data summarized by Chen

and Dudhia (2001), and their common soil type classification. Different soil types, predominantly expressed by different corn

sizes (corn diameter $D_p$) and pore space volume, available for soil water, can be classified according to the contribution of

sand (0.002 mm < $D_p$ < 2mm), silt (0.002 mm < $D_p$ < 0.050 mm) and clay ($D_p$ < 0.002 mm). Those mimic the potential of the

soil texture to fix water, against which the suction pressure of plants needs to work for extracting soil water. This determines

not only the *PWP* of the corresponding soil, i.e. the point at which a plant is unable to suck out any water of the soil pores, but

also the maximum amount of water a soil can hold against gravity (*FK* or $SWC_{v,max}$). The individual studies published refer to

different soil types, *PWPs* and $SWC_{v,max}$ that need to be considered and included in a more general parameterization. Here, we

use the contribution of individual soil types to the mixtures applied in the corresponding studies for calculating the *PWP* and





the $SWC_{v,max}$. In order to make different soil types and conditions comparable, all soil water describing parameters were
converted to $SWA_\%$ using $PWP$ and $nFK$.

As this study is part of a beech – silver fir research project in Southwestern Germany most figures shown are displayed with
two horizontal axes, (i) the reference $SWA_\%$ in percentage and (ii) an exemplary $SWC_v$ describing the one at the projects field
site in Freiamt, Black Forest, Southwestern Germany (Magh et al., 2018). This is representative for natural European beech
soil common in Central Europe (Gessler et al., 2007) with a $PWP$ of 4.7% and a $nFK$ of 31.2%. In order to make results
applicable to a wider range, any fitted equations take into account the soil properties and are provided for general conditions
as functions of (i) $SWA_\%$ (general reference) and of (ii) $SWC_v$ (requires adaptation to local conditions).

**2.3 Different fitting approaches and corresponding driving forces**

The effect of soil moisture on the emission of BVOCs is described by Guenther et al. (2006; 2012) using three-step pattern:
Assuming (1) no emission below the $PWP$, (2) a linear increase of emissions between $PWP$ ($\gamma_{SM} = 0$) and $PWP$+4 vol% of
$SWC_v$ ($\gamma_{SM} = 1$) and (3) a soil moisture independent emission above. This empirical parameterization was based on isoprene
emission measurements of Canadian black poplar (*Populus deltoides*) at the Biosphere 2 facility (Pegoraro et al., 2004a). A
different suggestion, exponential dependence of emission on $SWC_v$ was published recently by Genard-Zielinski et al. (2018)
for isoprene emissions too but for downy oak (*Quercus pubescens*) instead of black poplar.

For the overall soil moisture dependency of BVOC emissions at standard conditions ($T = 30°C$) several processes become
important depending on the molecular size of the compound, its production and storage behavior as well as its water solubility.

(i) Stomata controlled effect: If the size of the BVOC molecule does not allow penetration of the leaf or needle surface layer
the least barrier between plant and atmosphere are the stomata. Thus, the emission process is controlled by stomatal opening
behaviour of the plant species (Simpson et al., 1985). Low $SWA_\%$ and $SWC_v$ causes low $\psi$ - please note the negative scale,
which triggers closure of stomata in order to increase the resistance for water molecules between leaf and atmosphere, helping
to avoid loss of water by transpiration. Thus, $\gamma_{SM}$ alters according to hydraulic conductivity pattern (index 'h') given as

$$\gamma_{SM,h} = a_h + b_h \cdot SWA_\%/(c_h+SWA_\%) = a_h + b_h \cdot (SWC_v\text{-}PWP)/(c_h/nFK+SWC_v\text{-}PWP) \tag{4a}.$$

The curve starts at $a_h$, i.e. the residual emitted fraction at $PWP$ and increases to unity as $SWA_\%$ reaches 100% at $SWC_{v,max}$ ($FK$).
Both coefficients $b_h$ and $c_h$ determine the exact shape and slope of increase and are linked via

$$c_h = (b_h+SWC_v(PWP)\text{-}1)\cdot(SWC_{v,max}\text{-}PWP) \tag{4b}.$$

The shape is therefore different to the approach used in the MEGAN formulation (Guenther et al., 2006; 2012) as the effect
sets in already below $SWC_{max}$.



(ii) Diffusion controlled effect: If the emission may take place at least partially through the *cuticula*, perhaps for smaller chemical species, loss of plant water will cause a bending of the tissue surface. This will increase the cell pressure and influence conditions and forces of contained smaller BVOCs leading to diffusion towards the ambient, i.e. emission. This process acts

similar to biological growth processes and can be described by

$$\gamma_{SM,g} = \exp(-\exp(b_g \cdot \exp(1) \cdot (c_g - (SWC_v - PWP)/nFK)+1)) = \exp(-\exp(b_g \cdot \exp(1) \cdot (c_g - SWA_\%)+1)) \qquad (5).$$

Similar as above $b_g$ as well as $c_g$ represent curve shape parameters. The implicit $\gamma_{SM}$ value at *PWP*, i.e. $a_g$, is given by $\exp(-\exp(b_g \cdot \exp(1) \cdot (c_g)+1))$.

(iii) Water solubility and transport effect: Further important processes as water dependent productivity or transport (e.g. via

sap flow) will display either a linear behavior or a mixture of Eqs. 4 and 6. This depends on the limitations in the entire process chain from production, storage, potential transport and emission, which may be controlled by stomata opening or diffusion through the *cuticula*.

(iv) Plant defense or interaction effect: Finally, a rise of BVOC emissions shortly above the *PWP* is apparent in some MT but mainly SQT related studies. This may be explained by plant defensive strategies such as detoxification and reduction of radical

oxidative species (ROS) species, as most of these chemical species possess a high reactivity concerning ozone and radicals. The observations indicate an increase with decreasing $SWA_\%$ or $SWC_v$, until a rapid collapse close to *PWP*. We consider these observations to appear like a Gamma function type with a maximum near the *PWP*, i.e. $a \cdot (SWA_\% + b)^c \cdot \exp(-d \cdot (SWA_\% + b))$ with a set of parameters $a$, $b$, $c$ and $d$ characteristic for individual plant species. They may potentially reflect a species ability to respond to oxidative stress.

**2.4 Exemplary field studies: Application to ambient conditions**

**2.4.1 Plant nursery, Freiburg**

Eleven 8-10 years old *Fagus sylvatica* seedlings were studied in the plant nursery of the Chair of Ecosystem Physiology (48.014695° N, 7.832494° E) of the Albert Ludwigs Universität Freiburg with different soil water potentials, which were determined predawn and during daytime between July 19$^{th}$ and 31$^{st}$ 2018. Six trees served as control plants, which were watered

regularly, while rainwater was excluded by self-constructed roofs above the soil for the other five trees. Noteworthy, summer 2018 was extremely dry with less precipitation below 35 mm m$^{-2}$ during June and July, which is only one third of the 30 year average (90.5±40.5 mm m$^{-2}$, Freiburg airport, distance: ca. 900 m NNW, German Weather Service, [ftp://ftp-cdc.dwd.de/pub/CDC/observations_germany/climate/](ftp://ftp-cdc.dwd.de/pub/CDC/observations_germany/climate/) monthly/kl/) for those two months. Mean $SWC_v$ of control and stressed beeches were approximated by two methods, (i) water potential of the experimental plants was determined with a Scholander

pressure chamber (Scholander, 1966) at a cut branch predawn and around noon, and (ii) by $SWC_v$ measurements at a nearby field (ca. 600 m NW) with a similar soil structure (VWC, 10HS, Decagon, Washington, USA). BVOC emission with the





method described by Haberstroh et al. (2018). For this purpose, BVOCs emitted from beech leaves were collected during daytime using air- sampling tubes filled with Tenax (Gerstel). Analysis occurred on GC-MS system (GC model 7890 B, GC System, MS: 5975 C VL MSD with triple-Axis Detector, Agilent Technologies, Waldbronn) equipped with a multipurpose
sampler (MPS 2, Gerstel, Mülheim, Germany) (Magh et al., in prep.). More details on the analysis can be found here (Kleiber et al., 2017).

Based on the observed emission rates and corresponding forest air composition, relative changes in OH and ozone reactivity of the emission cocktail observed were derived as follows (Nölscher et al., 2014; Mogensen et al., 2015): The sum of the individual products of emission rates [molecules $m^{-2}$ $h^{-1}$], and their corresponding reaction rate constants [$cm^3$ molecule $s^{-1}$]
was calculated and compared to the related sum of the undisturbed reference trees. This includes the knowledge of a large set of compound reaction rates of which some are not obtainable in the available literature. Those have been approximated using structure activity relationship (SAR) based algorithms developed by Neeb (2000) and McGillen et al., (2011). In order to do so, the molecule of interest was split in its functional groups and the respective coefficients used for the estimate. As this approach is relative (i.e. disturbed to undisturbed trees), emissions of any vertical mixing is unimportant here, but will have to
be considered for the detailed atmospheric impact and its range. In this case, it is worth mentioning that forests extend usually to some tenths of kilometers in the Black Forest area and the area of emission can be easily traced back within the next kilometers in distance (unpublished data). This indicates the importance of BVOC emissions on the local atmospheric chemistry.

### 2.4.2 Black forest conditions at Freiamt

$SWC_v$ values were measured at five different depths (5, 10, 25, 50 and 75 cm) with a time resolution of 2h. Sensors were placed at different locations to test the heterogeneity, especially beneath different tree types, i.e. European beech (*Fagus sylvatica*) and Silver fir (*Abies alba*) and in between. For further information on measurements see Magh et al. (in prep.).

In order to approximate ambient $SWC_v$ effects on isoprene and terpene emissions, Eqs. 1, 3, 4 and 6 have been applied to meteorological measurements nearby and $SWC_v$.

## 3      Results

As indicated in section 2.3 we standardized and tested the influence of available soil water on emission rates of isoprene ($E_{isop}$), MT ($E_{MT}$) and SQT ($E_{SQT}$) using different hypotheses, a) a stepwise effect (Guenther et al., 2006), b) a growth rate like behavior (Eq. 5), a) hydraulic conductivity pattern (Eqs. 4a and b) as well as d) a stress defense response of $SWA_\%$ (lower x-axis) and for comparison the $SWC_v$ for a selected condition (upper horizontal axis).



### 3.1 Isoprene

Most studies on plant BVOC emissions affected by limited amounts of soil water investigated isoprene. A summary plot of individual studies rescaled to comparable conditions is provided in Fig. 1. In here the corresponding $SWA_\%$ conditions are plotted vs. $\gamma_{SM}$. A secondary horizontal axis on the top of Fig. 1 displays typical $SWC_v$ representative for Black Forest conditions at the Freiamt site (PWP = 4.7, nFK = 31.2%, soil: loamy sand; Magh et al., 2018). These conditions are similar to what has been found for European beech forest conditions elsewhere (Dalsgaard et al., 2011). Values of $\gamma_{SM}$ displayed were obtained from measured emission rates (Pegoraro et al., 2004a; 2004b; Brilli et al., 2007; Fortunati et al., 2008; Bourtsoukidis et al., 2014) and were divided by either (a) the corresponding measurements at well-watered standard conditions or (b) by literature values for the individual plant species. More information on the individual datasets and studies is provided in Table 2. Some values such as *PWP* and *nFK* are assumed according to the information provided by the authors of the corresponding studies. The figure clearly indicates that most of the studies focused on a specific smaller frame of the entire $SWA_\%$ range and only a single value has been measured below the *PWP*. This smaller range tended to yield a simplified description as for example the dataset of Pegoraro et al. (2004a), which was used by Guenther et al. (2006; 2012)(residual standard error (RSE) = 0.163, r² = 0.65). Testing different models for the dominating processes, the biological growth curve was found to fit the available datasets best (RSE = 0.06).

$$\gamma_{SM}(isop, fit, growth) = \exp\left(-exp\big((0.056 \pm 0.001) \cdot \exp(1) \cdot (-(2.3 \pm 1.5) - SWA_\%) + 1)\big)\right) \quad (6a)$$

$$= \exp\left(-exp\left(\tfrac{(0.056 \pm 0.001)}{nFK} \cdot \exp(1) \cdot \big(-(5.4 \pm 3.5) - (SWC_v - PWP)\big) + 1\right)\right) \quad (6b)$$

Apparently, limited soil water access seems to influence isoprene emissions predominantly by growth stress and less by stomatal opening, although growth stress and stomatal opening share similar features in a plot like Fig. 1 and only the entire $SWA_\%$ range allows discrimination between the two. Isoprene may diffuse out of the plant because of its smaller molecular size. Thus, a nearly closed stomata may not protect the plant from the release of isoprene because of enhanced cellular concentrations at reduced stomatal opening (Simpson et al., 1985; Fall et al., 1992).

### 3.2 Monoterpenes

The situation for monoterpenes (MT) is more complex as different MT isomers display a different behaviour with decreasing $SWA_\%$ (and $SWC_v$, Fig. 2). For example, in the case of European beech (*Fagus sylvestris*) the dominant MT sabinene (Moukhtar et al., 2006) declines drastically with reduced water availability, but limonene to a smaller extend, whereas trans-β-ocimene stays constant within the uncertainty limit (Rombach, 2018). Lüpke et al. (2017a) report declines in α- and β-pinene, limonene and myrcene emissions in response to increasing drought stress on Scots pine (*Pinus sylvestris*), while $\Delta^3$-carene emissions remained unaffected. An isotopic labeling test displayed a tendency of negatively influenced *de novo* production and contribution with declining $SWC_v$ and thus $SWA_\%$, while emission of stored monoterpenes was less affected. This may explain





the overall behavior of MT emission and propose the storage pools to determine the total amount of emissions. If so, any damage to cell walls or other plant structure elements, and extensive drought length will cause a) a significant decline in total emissions beyond the drought period on the long term, and b) the ability of the plant structure to recover, to form and store new MT as before. Individual MT structure behaviour however will depend on their detailed production pathway and storage location within the plant. Ormeno et al. (2007) describe an increase of MT emissions by rosemary (*Rosmarinus officialis*) at

reduced soil water supply under Mediterranean conditions. The individual contribution of different structures even display a further process to be noted: The relative contribution of α-pinene increased at reduction of $SWC_v$ from 21% to *PWP* and dropping thereafter similar to a stress response until cell damage was observed. A secondary contribution increase at very low $SWC_v$ values observed for Sweet chestnut (*Castanea sativa*; Lüpke et al., 2017b) may result from changes in very small emission amounts and appears as large uncertainty range awaiting further investigation. The remaining emission flux below

*PWP* may result from *de novo* production or from different storage locations, which are still functioning properly. Emissions of sabinene behave in a very similar way to the ones of α-pinene, while the fluxes of myrcene transiently increase around *PWP*. The opposite is true for cineole contribution that drastically declines with reducing $SWC_v$ and displays only at notable soil water presence. MT emissions of Aleppo pine (*Pinus halepensis*) display a pattern insignificantly influenced by $\gamma_{SM}$ vs. $SWC_v$ for α-pinene, $\Delta^3$-carene and linalool, while β-pinene and myrcene tend to increase. Total kermes oak (*Quercus coccifera*)

emissions of MT do not change significantly as well, but the contribution of α-pinene to total MT emissions enhances continuously even beyond the *PWP*. Therefore, the effect of individual MT structures depends on production rate as well as on storage pools and locations, which differ for different plant species. By changing the MT mixture plants may adapt to different stress conditions for improved defense (more details in section 3.5). However, the total MT emission effect of limited soil moisture seems to be more general.

A representative plot indicating the influence of $SWA_\%$ on total MT emissions is shown in Fig. 2. The set-up is identical to Fig. 1 for isoprene, i.e. the different studies included are marked with different symbols and colours. Lighter colours represent conditions after rewetting if performed as for instance for Scots pine (*Pinus sylvestris*) (Lüpke et al., 2017a). Lighter points indicate a severe plant structure damage and not entirely recovered plants reaching a lower emission intensity as has been observed in several studies. Excluding those and assuming a soil water supply above *PWP*, different fits can be applied to the

entire dataset. While all data scatter notably, the best performance is obtained with the hydraulic conductivity approach (residual standard error (RSE) = 0.02), which is stomata opening controlled. Eventually the biological growth fit (RSE = 0.14) works appropriately if the smallest values are neglected. However, the three-step approach by Guenther et al. (2006; 2012) fails (RSE = 0.69) to appropriately describe the observations. Due to different abilities to store MT as commonly denoted as offline and only temperature driven emissions the lower end of the fit displays a notable variance.

$$\gamma_{\text{SM}}(MT, fit, growth) = (0.22 \pm 0.05) + (0.78 \pm 0.05) \cdot \frac{(1.3 \pm 0.1) \cdot SWA_\%}{(33 \pm 3)\% + SWA_\%} \qquad (7a)$$





$$= (0.22 \pm 0.05) + (0.78 \pm 0.05) \cdot \frac{(1.3 \pm 0.1) \cdot (SWC_v - PWP)}{\frac{(33 \pm 3)\%}{nFK} + (SWC_v - PWP)} \qquad (7b)$$

$$\gamma_{SM}(MT, fit, growth) = \exp\left(-exp\left((0.019 \pm 0.004) \cdot \exp(1) \cdot (-(5.0 \pm 1.2) - SWA_\%) + 1)\right)\right) \qquad (8a)$$

$$= \exp\left(-exp\left((0.019 \pm 0.004) \cdot \exp(1) \cdot \left(-(5.0 \pm 1.2) - \frac{(SWC_v - PWP)}{nFK}\right) + 1\right)\right) \qquad (8b)$$

Note, the $SWA_\%$ effect is less punctual at a specific $SWA_\%$ than in the case of isoprene and the specific slope is soil, i.e. $nFK$

and $PWP$, dependent. The decline in emission strength gets already larger than 5% by 70±5% of the accessible $SWA_\%$,

depending on the fitting model chosen, not at about 6±2 in the case of isoprene as similarly found by Guenther et al. (2012).

However, the lower edge of the fitting is different but important especially for arid regions and their corresponding emissions.

On average about 22±5% of the total MT emissions remain unaffected by soil water content (at $PWP$).

### 3.3 Sesquiterpenes

Based on the molecular properties such as saturation vapour pressure (Seinfeld and Pandis, 2016), water solubility (Sander,

2015), and capability to be stored (Kosina et al., 2012), the effect of $SWA_\%$ on the emissions of sesquiterpenes (SQT, $C_{15}H_{24}$)

(Duhl et al., 2008) behaves the same way as seen for MT (Fig. 3), i.e. emissions are predominantly affected by plant hydraulic

conductivity (RSE = 0.153), which is controlled by stomatal opening. Applying the same approach as done for monoterpenes

above, yields the following results for the soil moisture effect on total sesquiterpene emission rates $\gamma_{SM}(SQT)$:

$$\gamma_{SM}(SQT, fit, hyd.) = (0.22 \pm 0.04) + (0.78 \pm 0.04) \cdot \frac{(1.75 \pm 0.5) \cdot SWA_\%}{(81 \pm 57)\% + SWA_\%} \qquad (9a)$$

$$= (0.22 \pm 0.05) + (0.78 \pm 0.05) \cdot \frac{(1.75 \pm 0.5) \cdot (SWC_v - PWP)}{\frac{(81 \pm 57)\%}{nFK} + (SWC_v - PWP)} \qquad (9b)$$

The onset of emission reduction with decreasing $SWA_\%$ sets in even earlier compared to MTs (Rombach, 2018), as indicated

by the different slopes in Fig. 3 and in Eqs. (9a and 9b). However, individual structures reveal a large scattering at lower $SWA_\%$

and thus $SWC_v$ values (<10%) that is supposed to result from plant defensive strategies/mechanisms and perhaps from damage

to cell walls and membranes.

### 3.4 Potential additional effects near the wilting point for terpenes

For some plant species e.g. *Cistus albidus* and rosemary (*Rosmarinus officialis*), the effects of limited access to soil water on

MT and SQT emission fluxes reveal a second maximum other than the one at $\gamma_{SM} = 1$, appearing near the $PWP$ (Figs. 4, S3-

S8). Highly temperature- and drought stress adapted species with the ability of to produce large amounts of terpenoids (e.g.





*Cistus ladanifer* (Haberstroh et al., 2018) and *Rosmarinus officialis* (Ormeno et al., 2007)) display a second maximum close
to *PWP* not only for some of the MT (*Cistus ladanifer*: d-limonene, myrcene and trans-β-ocimene (Haberstroh et al., 2018);
*Rosmarinus off.*: α-pinene and β-myrcene (Ormeno et al., 2007)) but also for some of the SQT (*Cistus ladanifer*: α-cubebene,
α-farnesene, ent-16-kaurene and β-chamigrene (Haberstroh et al., 2018); *Rosmarinus off.*: Δ-cardinene and α-zingiberene
(Ormeno et al., 2007), Fig. 4) and even the sum of all. This is apparent for instance in the studies of Ormeno et al. (2007) and

Haberstroh et al. (2018) of a broader range of plant species. As can be seen in Figs. S3-S8 (supporting online information),
there is a statistically significant difference between individual chemical structures for different plant species, indicating a
specifically adopted plant species. If those plots are investigated in more detail it is apparent that for example the monoterpene
α-pinene increases with decreasing $SWA_\%$ for *Rosmarinus officialis* (S6) and *Quercus coccifera* (S5), but persists for *Pinus
halepensis* (S4) (Ormeno et al., 2007). The emission of the monoterpene β-pinene is declining for *Cistus albidus* (S3), stays

put for *Rosmarinus officialis* (S6) and enhances for *Pinus halepensis* and *Quercus coccifera*. Similar observations can be made
for SQT emissions with *SM* values near the *PWP*: Δ-cadinene emissions increase with decreasing *SM*, while α-zingiberene
emission decline for *Rosmarinus officialis*. Haberstroh et al. (2018) monitored decreasing emission fluxes for all the individual
chemical SQT structures for *Cistus ladanifer* and *Quercus robur* in a very narrow SM range ($SWA_\%$ = ca. 0-6 %). This is
opposite to the behaviour of *Cistus albidus* (Ormeno et al., 2007). Δ-germacrene emissions tend to increase near the *PWP* for

*Cistus albidus*, *Quercus coccifera* and *Rosmarinus officialis* (Figs. S3, S5 and S6).

The enhanced emitted chemical species are known to possess a substantially higher reactivity with respect to ozone, which is
usually elevated at drier conditions on longer time scales because of enhanced production rates and accumulation (see section
3.7). For example α-pinene possesses an $O_3$ reaction rate constant ($k_{O3}$) of $8.66 \times 10^{-17}$ cm$^{-3}$ molecule$^{-1}$ s$^{-1}$, compared to the ones
of β-pinene and $\Delta^3$-carene of $1.5 \times 10^{-17}$ and $3.7 \times 10^{-17}$ cm$^3$ molecule$^{-1}$ h$^{-1}$, respectively (Atkinson et al., 1990). With respect to

the OH reaction, β-pinene reacts fasted ($k_{OH}$ = $7.89 \times 10^{-11}$ cm$^3$ molecule$^{-1}$ h$^{-1}$), followed by α-pinene and $\Delta^3$-carene with
corresponding rate constant $k_{OH}$ of $5.37 \times 10^{-11}$ and $0.88 \times 10^{-11}$ cm$^3$ molecule$^{-1}$ h$^{-1}$, (Atkinson and Arey, 2003). This can be
extended for other monoterpenes if more data especially on d-limonene, terpinolene and other very reactive MTs would be
available. However, the number of data for individual species is quite limited and depends on different plant ages, which does
not allow deriving a trustworthy equation at this stage. Sometimes the number of data points is equal to the number of

parameters to constrain. The situation looks similar for SM effects on SQT emission rates. Consider e.g. the bottom plot for
SQT in Fig. 4, displaying allo-aromadendrene, Δ-germacrene, Δ-cadinene and α-zingiberene. Reaction rate constants $k_{O3}$ with
respect to ozone are $6.5 \times 10^{-16}$, unknown, $3.2 \times 10^{-16}$ and unknown, respectively. The corresponding $k_{OH}$ for OH reactions are
$1.5 \times 10^{-10}$ cm$^3$ molecule$^{-1}$ h$^{-1}$ for aromadendrene, with the further ones not accessible based on experiments. However, structure-
activity relationships such as EPI Suite (2018), indicate constants of ca $10^{-10}$ cm$^3$ molecule$^{-1}$ h$^{-1}$. Thus, we assume this

secondary maximum to occur as detoxification and defensive strategy of the plants (Kaurinovic et al., 2010; Höferl et al.,
2015). Because of notable variations in measurements this effect is not significant for all the Mediterranean species, but for





the *Cistus ladanifer* and *Rosmarinus officialis* measurements. To incorporate this feature near the *PWP* exemplarily, which tends to be linked to higher ozone concentrations, this secondary maximum can be taken into account using a Gamma function term added to the hydraulic conductivity curve (RSE = 0.147) in Eqs. 9a and 9b:

$$\gamma_{SM}(SQT, fit, hyd.)^* = (0.22 \pm 0.04) + (0.78 \pm 0.04) \cdot \frac{(1.75 \pm 0.5) \cdot SWA_\%}{(81 \pm 57)\% + SWA_\%}$$
$$+ \text{a} \cdot (SWA_\% + b)^c \cdot exp\big(-d \cdot (SWA_\% + b)\big) \tag{9a}$$

$$\gamma_{SM}(SQT, fit, hyd.)^* = (0.22 \pm 0.04) + (0.78 \pm 0.04) \cdot \frac{(1.75 \pm 0.5) \cdot (SWC_v - PWP)}{\frac{(81 \pm 57)\%}{nFK} + (SWC_v - PWP)}$$
$$+ \text{a} \cdot \left(\frac{SWC_v - PWP}{nFK} + b\right)^c \cdot exp\left(-d \cdot \left(\frac{SWC_v - PWP}{nFK} + b\right)\right) \tag{9b}$$

*a*, *b*, *c* and *d* state parameters for the individual response of plant species to severe drought stress Gamma function) and they need to be adapted to the exact plant or ecosystems response studied on the local scale. An example is shown by the green line in Fig. 3 (applied parameters in here for *Cistus ladanifer* and *Rosmarinus officialis*: a = 0.4, b = 1.57, c = 0.6, d = 1). It is of interest that different structures seem to behave differently according to their historical stress adaptation as presented by Fortunati et al. (2008) and Lüpke et al. (2016). On the contrary to the hydraulic conductivity related description, the stepwise approach (Guenther et al., 2006) fails in reproducing the declining pattern accurately (RSE = 0.895).

## 3.5 Further BVOC emission rates

In addition to isoprene, MT and SQT, other VOCs are released by plants in notable amounts, but for which the impact of soil moisture is even less studied. Because of the huge number of different oxidized species and related publications including those within the parameterization would extend this study largely. A snapshot of individual correlation coefficients of different compound groups and species can be seen in Figure S1. A non-negligible role of those larger VOCs seems plausible, either via formation pathway or oxidation processes within the plant. Those species supply further molecules that can buffer oxidative stresses onto the plants internal processes. Additionally, they have substantial implications on especially the OH reactivity (Atkinson et al., 1990; Atkinson and Arey, 2003; Neeb, 2000; Münz, 2010; Seinfeld and Pandis, 2016, US EPA, 2018). Overall, there is a tendency for an increased total BVOC emission under drought stressed conditions, but there is a large scatter of individual species reaction. However, the implications for OH reactivity and thus atmospheric chemistry are of relevance as this parameter describes the plants ability to still counteract oxidative damages by BVOC emissions.





### 3.6 Effects on reactivity

As chemical oxidation and ROS cause substantial damage to plants (Loreto and Velikova, 2001; Oikeawa et al., 2013)
especially at stressed conditions such as at warm and dry conditions, any strategy of plants to counteract or detoxify will be
beneficial for survival and plant competition. An indication for this may be seen in the local emission increase of SQT near
*PWP* for some plant species. Saunier et al. (2017) found a stable ratio of catabolic to anabolic BVOC emissions, i.e. emission
fluxes related to oxidation stress and to non-oxidative stress production, respectively, during recurring drought periods for
downy oak (*Quercus pubescens*) at Mediterranean conditions. This is different for different plant types and their adaptation to
stress conditions (Niinemets, 2010a; 2010b), i.e. the need to compensate oxidative stress for survival. Looking at the pattern
of absolute and relative MT and SQT emissions with declining $SWA_\%$ values, it becomes obvious that more reactive structures
tend to increase until plant cell damages occur (Ormeno et al., 2007; Bourtsoukidis et al., 2012; Nölscher et al., 2016).
Especially SQT emissions displaying this transient peak around *PWP* cause a remarkable change. For example reducing the
$SWC_v$ for European beech by approximately 3 vol% close to *PWP* increases the total emission of SQTs by 170%. The individual
SQT species, however, show different patterns (Table 2). While emission of junipene declines (-50%), the one of longifolene
stays constant within the uncertainty limits and the corresponding fluxes of isolongifolene, α-bergamotene and α-farnesene
increase drastically (+50%, +500% and +700%). Other VOCs analyzed (total: 81 VOCs; Rombach, 2018) operate in various
ways: sabinene emission nearly vanishes at *PWP*. On the contrary, emission of α-terpineol increased. Overall, the total
emissions of VOCs and C increased tentatively (Fig. 5). This caused total OH reactivity and especially ozone reactivity to
tentatively enhance (OH: $+39^{+88}_{-46}$% and O$_3$: $+131^{+265}_{-104}$%) at a $SWC_v$ change of approximately 3 vol%. In this way the total
amount, the changing composition and reactivity provide a tool for plants in this example for European beech to counteract
oxidative stress and damages. The median change in OH reactivity corresponds nicely to the increase in total carbon emitted,
as most organic species react with OH. However, the change in ozone reactivity is primarily related to the changes in individual
SQT emissions. The large uncertainty in this context results from partially unknown reaction rate constants estimated either
from similarity or from structure activity relationships (SAR) (Neeb, 2000; McGillen et al., 2011). Since the storage pools
especially of larger terpenes will be depleted with prolonged duration of drought, this effect will weaken over time.

### 3.7 Application to atmospheric conditions: Ambient $SWC_v$ and its effect on emission rates at the reference site

In order to test the impact of the described effects of $SWA_\%$ and thus $SWC_v$ on isoprenoid emissions we apply the following
emission flux parameterizations for isoprenoids from European beech and silver fir, which were quantified in earlier studies
in this region excluding the $SWA_\%$ effect (Moukhtar et al., 2005; 2006; Šimpraga et al., 2011; Bonn et al., 2017):

$$E_{isop}(Fs) = 0 \text{ µg g(dw)}^{-1} \text{ h}^{-1} *C_T*C_L \tag{10a}$$

$$E_{MT}(Fs) = 43.5 \text{ µg g(dw)}^{-1} \text{ h}^{-1} *C_T*C_L \tag{10b}$$



$E_{SQT}(Fs) = 0.04$ µg g(dw)$^{-1}$ h$^{-1}$ *exp(0.11 °C$^{-1}$*($T_L$-30°C)) (10c)

$E_{isop}(Aa) = 0.038$ µg g(dw)$^{-1}$ h$^{-1}$ *$C_T$*$C_L$ (10d)

$E_{MT}(Aa) = 28.8$ µg g(dw)$^{-1}$ h$^{-1}$ *(0.71*exp(0.14 °C$^{-1}$*($T_L$-30°C))+0.29*$C_T$*$C_L$) (10e)

$E_{SQT}(Aa) = 0.13$ µg g(dw)$^{-1}$ h$^{-1}$ *exp(0.04 °C$^{-1}$*($T_L$-30°C)). (10f)

*SM* effects on specific emission rates have been calculated by including and excluding the individual $\gamma_{SM}$ factors as multiplicand on the results. In Eqs. 10a-10f $C_T$ and $C_L$ abbreviate the temperature and light scaling factors of Guenther et al. (1995), as well as $T_L$ the temperature of the needle or leaf surface in °C.

The base effect of soil water on isoprenoid emission rates can be exemplarily shown assuming a temperature of 30 °C, i.e. standard conditions (see Fig. S2 in supporting online information). The available soil water for a plant has different impacts at distinct *SWA%* values. At *SWA%* (i.e. *SWC$_v$* (Freiamt) = 20%) of 35% isoprene emission rates display no decline, while the corresponding ones of MT and SQT reduce by 15 and 25%, respectively. Assuming an *SWA%* of 10% (*SWC$_v$* = 10% with a *PWP* of 4.8% at Freiamt), isoprene, MT and SQT emission rates represent only 95, 50 and 40% of the standard rates. Therefore, the amount of soil water has strongest effects on the larger terpenes shrinking with molecular size. Isoprene emissions will stay unaffected longer than MT and will keep ozone production high if isoprene is the dominant class of isoprenoids released by the ecosystems in the nearby region. Depending on the tree specific emission – i.e. on- and offline including temperature effects – different locations of emission optima or maxima are found (Fig. 6). The predominantly *de novo* emitting European beech will emit highest close to 30°C, while the maxima of silver fir emission rates are shifted to larger temperatures. Similarly the effects of changing *SM* will occur at the very same temperatures.

The overall impact of variable *SM* pattern on isoprenoid emissions was investigated using an exemplary *SWC$_v$* data at five different depths. Those measurements were located within the reference stand of a European beech (*Fagus sylvestris, Fs*) and silver fir (*Abies alba, Aa*) of the BuTaKli project in Freiamt within the Black Forest (Southwestern Germany) region (Magh et al., 2018). In general, the uppermost layer is drying out fastest especially during summertime, while lower layers, i.e. below 30 cm, retain a higher *SWC$_v$* for a longer time period. This implies different effects on the tree species with different rooting depths and thus access to soil water. Regarding the effect of *SM* on isoprenoid emissions - i.e. the g factor - the $g_{SM}$ values for one vegetation period (during 2016) are shown in Fig. 7. Individual plots represent the *SWC$_v$* emission effects of different compounds or compound classes and different colours illustrate the *SWC$_v$* based values for three different depths. Apparently during summertime, when the evapotranspiration flux and the emission rates are highest due to elevated temperatures and radiation intensity, the effect is most intense. During the rather well watered summer 2016 $\gamma_{SM}$(*isop*) caused reduction by almost 40% for more shallow rooting species and by 20 to 30% for trees with access to deeper soil layers (60 cm in depth).





$\gamma_{SM}(MT)$ declined to nearly 55% on average with short time access to more water for deeper roots and $\gamma_{SM}(SQT)$ to nearly 60% of emission rates.

During 2017 with an improved roof cover, the differences between different soil depths became more obvious, and they exceed
effect changed by 10% or more between 40 and 60 cm in depth (not shown, (Magh et al., in prep). Those effects are estimated to intensify remarkably during extensive drought periods such as occurred in summer 2018, of which no dataset is available. This is going to take down most of the BVOC emission rates with progressing drought period time, depending on the tree species rooting depth and soil water access. Therefore, the total sum of isoprenoid emissions of a specific forest ecosystem will be reflected by the individual contribution of tree species and its specific emission pattern, and the sum of *SM* affects the
drought sensitivity of the ecosystem.

## 4      Conclusions

In this study we derived three sets of parameterizations for the influence of soil moisture on isoprenoid emission rates. Those depend on the details of soil characteristics, plant properties of on- and offline production of isoprenoids and potential defensive plant feedback mechanisms on oxidative stress. The effect on isoprene emissions acts similar as a small molecule able to
penetrate the plant surface acting like a biological growth stress. While the cuticula provides some minor resistance it cannot hinder isoprene from penetration at nearly closed stomata. Thus, the present description allows minor isoprene emissions below the *PWP*. The influence of soil water access on larger isoprenoid emission occurs on a notably larger range of $SWA_\%$ as a function of stomata opening (hydraulic conductivity pattern). Depending on the specific tree species some molecular structures of terpenes e.g. the more reactive ones like α-pinene and β-myrcene (Haberstroh et al., 2018; Ormeno et al., 2007) and α-
farnesene (Duhl et al., 2008; Haberstroh et al.,2018) display an elevated relative terpene flux contribution at lower soil moisture (Figs. S3-S8), indicating a defensive strategy of plants against oxidative stress at lower soil water concentrations, thus warmer conditions and in total critical for the plants survival. This is plant species dependent and can be incorporated by a secondary maximum close to the permanent wilting point. The named $SWA_\%$ impact on isoprenoid emission rates is expected to induce several feedback processes in the atmosphere and climate system including local tropospheric ozone production and sink
(Seinfeld and Pandis, 2016), new particle formation (Bonn et al., 2014; Bonn 2014) and growth as well as cloud condensation nuclei (Bonn 2014; Seinfeld and Pandis, 2016).With these findings we conclude that more studies on available soil water dependence of emissions of higher terpenes, oxygenates and follow-up processes, are essentially needed for different environments to realistically describe the natural feedback processes of forests and ecosystems in changing climate conditions and the anthropogenic impact on these. Because of that, the $SWA_\%$ concepts described should definitely be included in
corresponding global atmospheric chemistry transport models in order to cover not only plant response to the expected increase in drought periods but to future development of biosphere-atmosphere processes, resulting in adaptation or extinction in a changing climate.

**Supplementary Materials:** Supplementary material is available as an additional file.

**Author Contributions:** B.B. was responsible for the methodology, analysis and writing. R.-K.M. conducted the *SWC$_v$* measurements and
soil texture analysis, contributed to the data analysis and parameterization approach and was responsible for the soil water concepts. The
formal analysis of Fagus sylvatica BVOC emissions and water potential measurements at the campus field site was done by J.R. J.K.
contributed by conceptualization of the Fagus sylvatica campus field site experiment, discussion about accurate plant physiology process
description as well as review and editing.

**Funding:** This research was funded by German Waldklimafond project BuTaKli, no. 100261371.

**Acknowledgments:** Funding within the German Waldklimafond project BuTaKli (no. 100261371) is kindly acknowledged. Highly
acknowledged is the supply of *SWC$_v$* data for Campus Flugplatz soil texture by Angelika Küberts and Simon Haberstrohs support with
further information and data on the Haberstroh et al. (2018) measurements. Thanks go to the supporting staff and colleagues at Freiburg
University and to all partners within the BuTaKli project for their nice support and exchange of information and data, too. We thank the R
Development Core Team for providing version 3.5.1 with its individual tool packages (R Core Team, 2017), which were used for conducting
this study.

**Conflicts of Interest:** The authors declare no conflict of interest.

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



**Table 1. Overview of abbreviated terms and parameter names**

| Abbreviation | name | Abbreviation | name |
|---|---|---|---|
| $a, b, c$ | fitting parameters, indices 'g' and 'h' reflect growth and hydraulic conductivity based fits | $NO_x$ | nitrogen oxides (=$NO$+$NO_2$) |
| $\beta$ | temperature dependent increase of emission rates | $PAR$ | photosynthetically active radiation |
| $CCN$ | cloud condensation nuclei | $PWP$ | permanent wilting point |
| $E$ | emission rate of compound (class) [mg g(dw)$^{-1}$ h$^{-1}$] | $ROS$ | reactive oxidation species |
| $E_0$ | emission rate at standard cond. | $RSE$ | residual standard error |
| $FASW_\%$ | fraction of available soil water [%] | $SAR$ | structure activity relationships |
| $FK$ | field capacity | $SM$ | soil moisture |
| $\gamma_T, \gamma_L, \gamma_{CO2,} \gamma_{SM}$ | emission regulating factor for the impact of T, light, $CO_2$ and soil moisture | $SQT$ | sesquiterpenes ($C_{15}H_{24}$) |
| $\gamma_{SM,g}, \gamma_{SM,h}$ | $\gamma_{SM}$ described by growth or hydraulic conductivity curve | $SWA_\%, SWA_v$ | soil water availability, percentile [%] and volumetric [v] |
| $k_{OH}, k_{O3}$ | reaction rate constants with OH and ozone | $SWC_v, SWC_m$ | Soil water content volumetric [vol %] and by mass [mass %] |
| $MT$ | monoterpenes ($C_{10}H_{15}$) | $VPD$ | vapour pressure deficit |
| $nFK$ | net field capacity | $\psi, \psi_{pd}$ | matrix or water potential [MPa] |






**Table 2.** Overview of studies, tree species and corresponding conditions as well as soil water parameters used for deriving the $g_{SWC}$ parameterization for isoprene, MT and SQT emissions. References to the individual studies are listed in the final column. Soil water status parameters include the fraction of available soil water ($FASW$ in %), leaf water content, available soil water ($SWA_\%$ in %), soil water content by volume ($SWC_v$), by mass ($SWC_m$) and by the ratio water volume per soil mass ($SWC_{vm}$), stem diameter and water potential during daytime ($\psi$ in MPa) and predawn ($\psi_{PD}$ in MPa),

| Tree species | | Soil type | PWP ass. | VOC species classified | Parameter describing | Rewa- | environment | Reference |
|---|---|---|---|---|---|---|---|---|
| Lat. Name | engl. name | | [$m^3\ m^{-3}$] | | water status | tering | | |
| *Castanea sativa* | sw. chestnut | 70% sand, 30% humus? | 2.8 | different MTs | $SWC_v$ | no | plant chamber | Lüpke et al., 2017a |
| *Cistus albidus* | rockrose | calcareous soil | 6.7 | MTs, SQTs | $\psi$, $SWC_{vm}$ | no | pots, greenhouse? | Ormeno et al., 2007 |
| *Cistus ladanifer* | common gum cistus | clay loam | 10.2 | Ox. MTs, MTs, SQTs, diterpenes | $SWC_v$, $\psi$ | no | ambient | Haberstroh et al, 2018 |
| *Cistus sinensis* | orange | clay loam | 10.9 | trans-β-ocimene (MT), β-caroyphyllene (SQTs) | $SWC_v$, $\psi$ | yes | pot, greenhouse | Hansen and Seufert, 1999 |
| *Fagus sylvatica* | European beech | CPS+fertilizer: 100% org? | 6.0 | total MTs | stem diameter | yes | growth room | Simpraga et al., 2011 |
| *Fagus sylvatica* | European beech | Bedrock? | 9.0±1.0 | Isoprene, MTs, SQTs + 70 VOCs | Leaf water content, $\psi_{PD}$ and $\psi$, $SWC_v$ nearby | no | ambient | Rombach, 2018 |
| *Pinus sylvestris* | pine | 70% sand, 30% humus | 2.8 | 1,8-cineole, different MTs | $SWC_v$ | yes | plant chamber | Lüpke et al., 2017b |
| *Populus alba* | white poplar | CPS: 100% org | 6.0 | Isoprene | $SWA_\%$ | yes | growth chamber | Brilli et al., 2007 |
| *Populus deltoids* | cottonwood | 60% bare soil, 40% organic | 7.4 | Isoprene | $SWC_v$, $\psi_{PD}$ | no | greenhouse | Pegoraro et al., 2004a |
| *Populus nigra* | black poplar | 50% org., 50% sand | 3.5 | Isoprene | $FASW_\%$ | yes | growth chamber | Fortunati et al., 2008 |
| *Prunus serotina* | cherry | 70% , 30% | 4.7 | MTs, SQTs, selected OVOCs | $SWA_\%$ | no | greenhouse | Bourtsoukidis et al., 2014 |
| *Rosmarinus off.* | rosemary | calcareous soil | 6.7 | MTs, SQTs | $\psi$, $SWC_\%$ | no | pots, greenhouse? | Ormeno et al., 2007 |
| *Quercus ilex* | oak | Silty clay loam | 13.6 | different MTs | $\psi$ | yes | ambient | Lavoir et al., 2009 |
| *Quercus robur* | oak | 70% , 30% | 4.7 | MTs, SQTs, selected OVOCs | $SWA_\%$ | no | greenhouse | Bourtsoukidis et al., 2014 |
| *Quercus virginiana Mill.* | live oak | CPS: 100% org.? | 5.0 | Isoprene | $SWC_v$, $\psi_{PD}$ | no | phytotron | Pegoraro et al., 2004b |

*CPS: commercial potting soil (e.g. by Agrofino, Miracle Grow) including or plus fertilizer (Basacot)*



**Table 3. Change for individual mono- and sesquiterpene species emission fluxes of *Fagus sylvatica* for two *SM* conditions (Rombach, 2018). Because of not normal distributed data, values are stated as median and 25th (lower) and 75th (upper) percentiles.**

| *Species* | *SWA% [%]* | | *Total amount [μg g(dw) h]* | | *Rel. amount [%]* | |
|---|---|---|---|---|---|---|
| | *A* | *B* | *A* | *B* | *A* | *B* |
| ***monoterpenes*** | $6_4^8$ | $3_1^5$ | $11.7_{2.9}^{35.7}$ | $0.8_{0.4}^{2.4}$ | **100** | |
| Sabinene | $6_4^8$ | $3_1^5$ | $11.2_{2.6}^{21.9}$ | $0.2_{0.1}^{3.0}$ | $91_{79}^{95}$ | $37_{23}^{75}$ |
| Limonene | $6_4^8$ | $3_1^5$ | $0.3_{0.1}^{1.3}$ | $0.1_{0.1}^{0.3}$ | $5_3^9$ | $12_8^{19}$ |
| trans-β-ocimene | $6_4^8$ | $3_1^5$ | $0.2_{0.0}^{1.2}$ | $0.2_{0.1}^{0.9}$ | $1_0^{10}$ | $39_{10}^{64}$ |
| ***sesquiterpenes*** | $6_4^8$ | $3_1^5$ | $10.43_{2.88}^{17.23}$ | $27.65_{12.12}^{42.27}$ | **100** | |
| isolongifolene | $6_4^8$ | $3_1^5$ | $9.10_{2.22}^{15.57}$ | $12.43_{6.60}^{30.17}$ | $88_{63}^{97}$ | $82_{35}^{97}$ |
| longifolene | $6_4^8$ | $3_1^5$ | $0.39_{0.15}^{0.79}$ | $0.30_{0.20}^{1.56}$ | $4_2^{12}$ | $1_1^6$ |
| α-farnesene | $6_4^8$ | $3_1^5$ | $0.34_{0.05}^{0.57}$ | $2.64_{0.21}^{6.56}$ | $3_1^7$ | $9_1^{49}$ |
| Junipene | $6_4^8$ | $3_1^5$ | $0.12_{0.04}^{0.15}$ | $0.06_{0.04}^{0.50}$ | $1_1^2$ | $1_0^6$ |
| α-bergamotene | $6_4^8$ | $3_1^5$ | $0.05_{0.02}^{0.07}$ | $0.29_{0.05}^{0.77}$ | $0_0^1$ | $1_0^6$ |






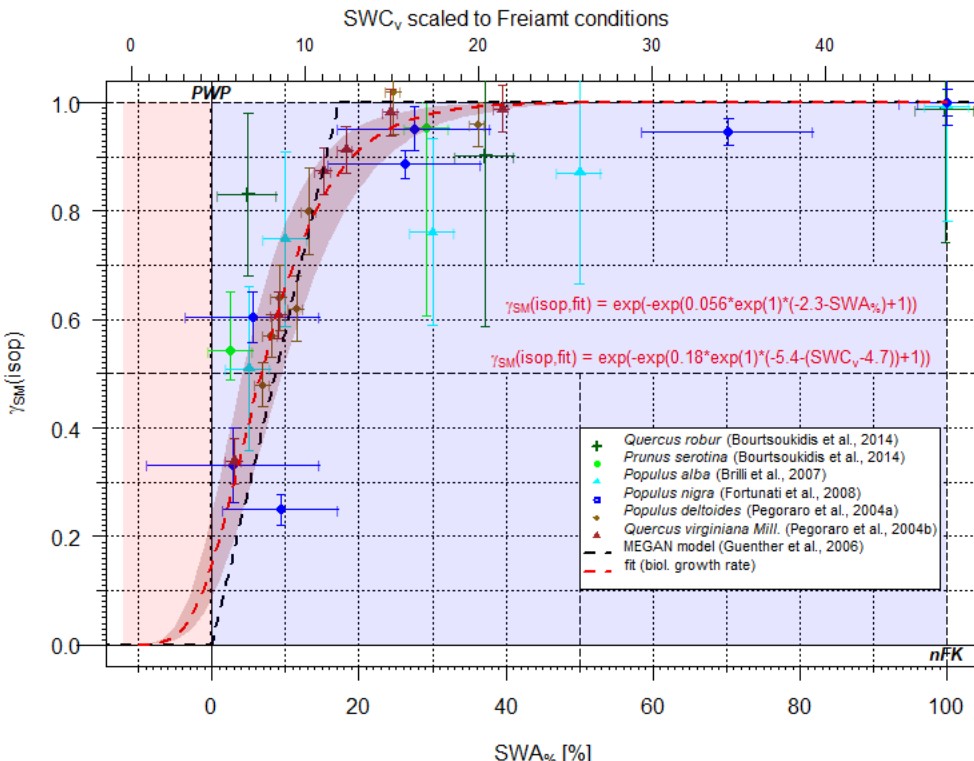

**Figure 1: Scaling parameter γ$_{SM}$(isop) for the effect of soil moisture on isoprene emissions as a function of SWA$_\%$ (lower x-axis) and for comparison the *SWC$_v$* for a selected condition (upper horizontal axis). Data points represent observations, and different colours refer to different studies. The black dashed line displays the approach by Guenther et al. (2006; 2012), the red one the present form.**

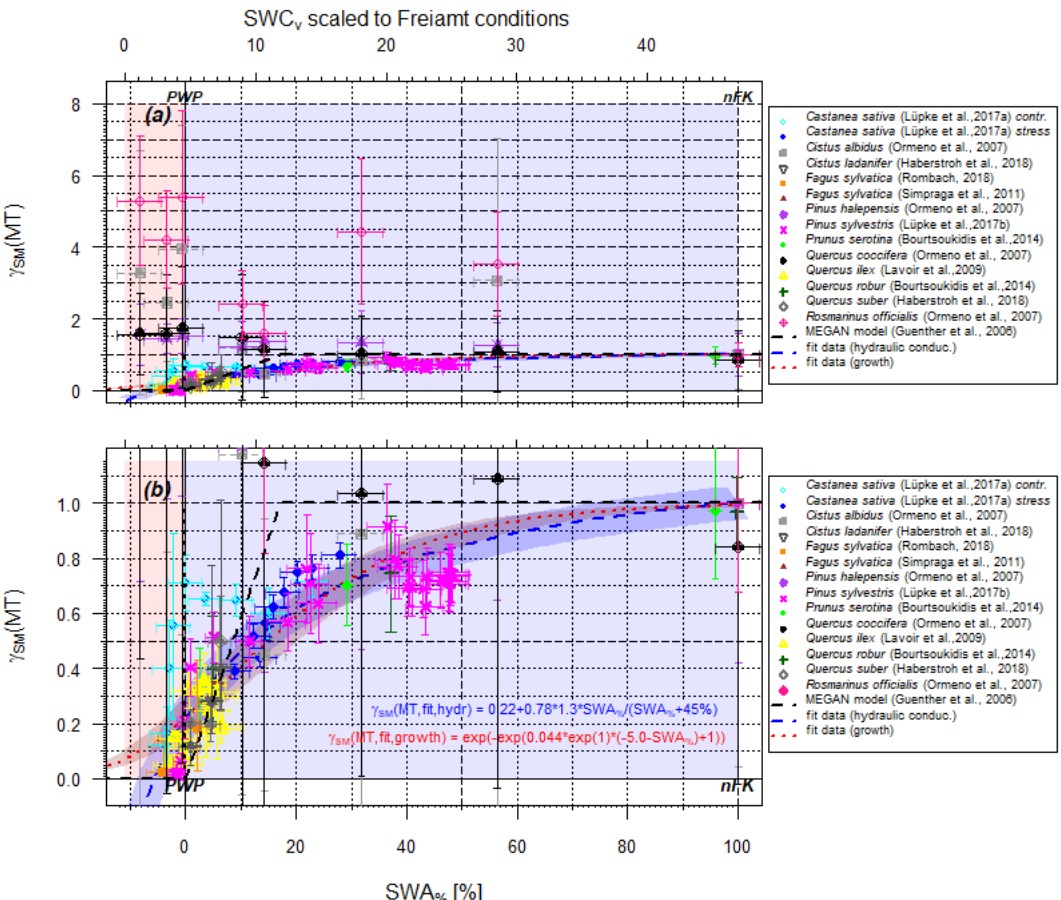

Figure 2: Scaling parameter $\gamma_{SM}(MT)$ for the effect of soil moisture on total MT emissions as a function of $SWA_\%$ (lower x-axis) and for comparison the $SWC_v$ for a selected condition (upper horizontal axis). The upper plot (a) displays the full data range, while the lower plot (b) zooms vertical area between 0 and 1.2. Data points represent observations, and different colours refer to different studies. The black dashed line displays the approach by Guenther et al. (2006; 2012), the red one the present form using hydraulic conductivity as driving force and the darkred one the biological growth stress as driving force.


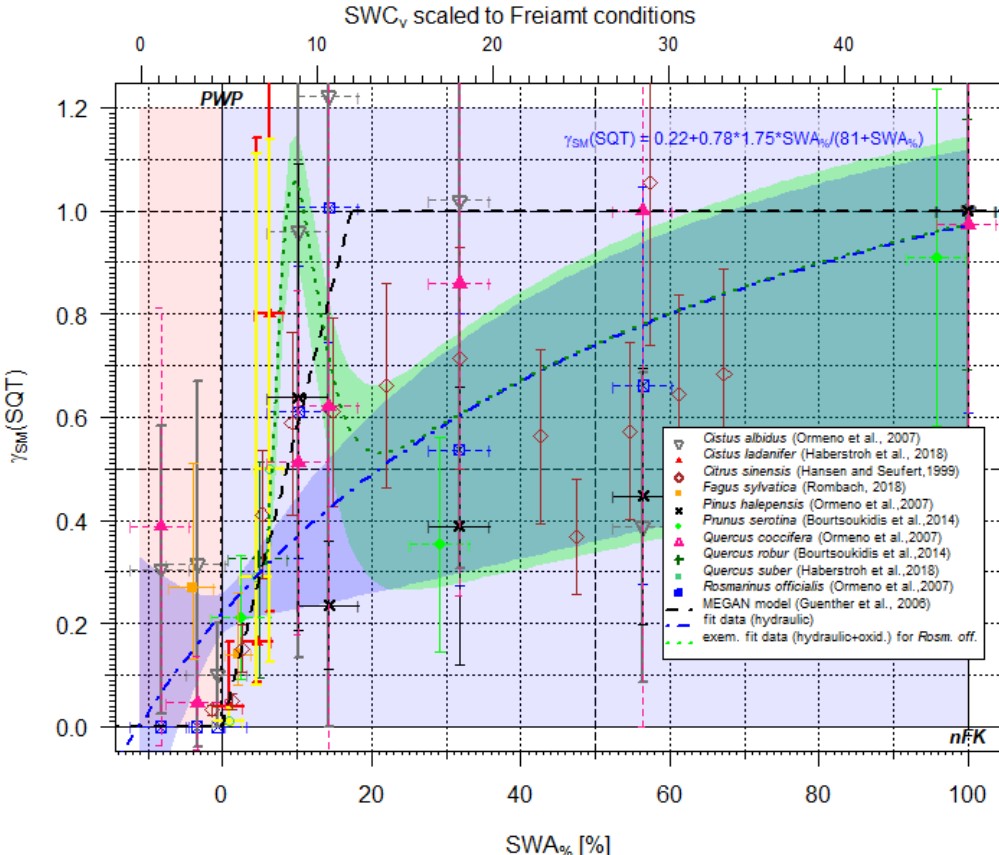

Figure 3: Scaling parameter $\gamma_{SM}(SQT)$ for the effect of soil moisture on SQT emissions as a function of $SWA_\%$ (lower x-axis) and for comparison the $SWC_v$ for a selected condition (upper horizontal axis). Data points represent observations, different colours refer to different studies. The black dashed line displays the approach by Guenther et al. (2006; 2012), the blue one the present form. The green pointed line represents an exemplary fit to $\gamma_{SM}(SQT)$ based on *Cistus* and *Rosmarinus officialis* data.





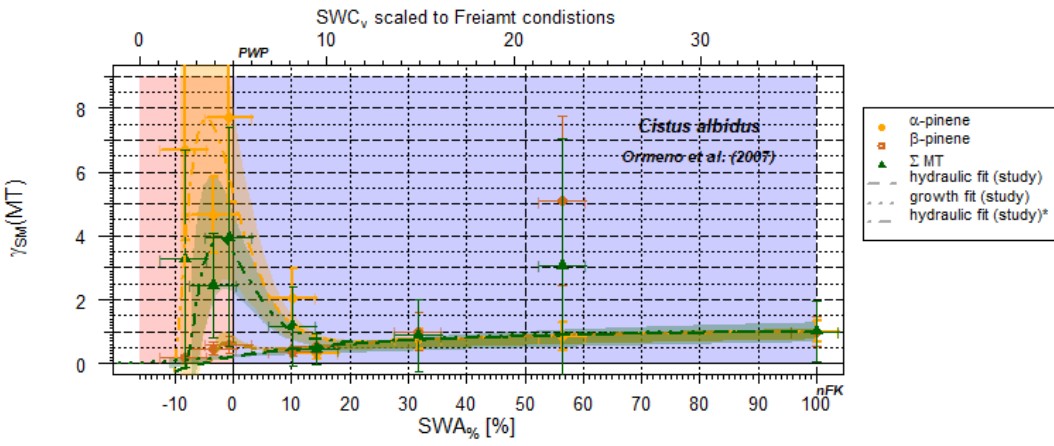

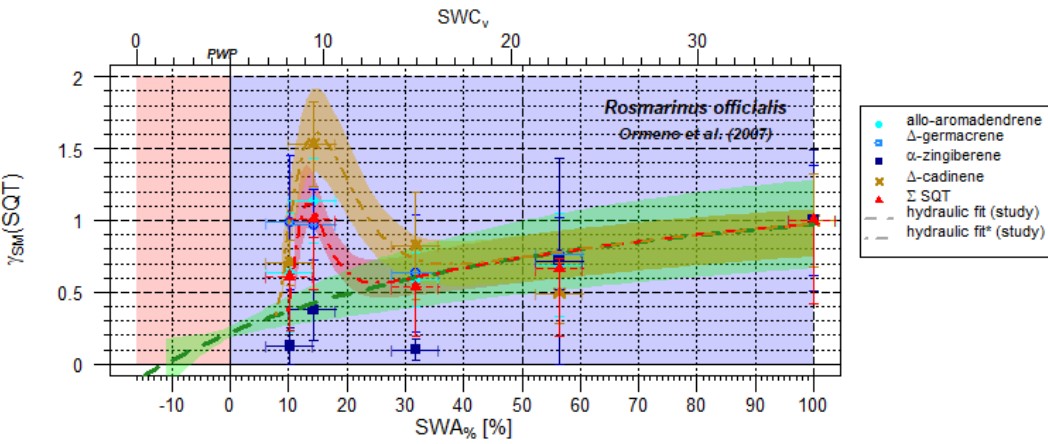

**Figure 4: Exemplary scaling parameter γ$_{SM}$ for the effect of soil moisture with a maximum at *PWP* on individual and total terpene emissions as a function of *SWA*$_\%$ (lower x-axis), and for comparing the *SWC*$_v$ at Freiamt soil conditions (upper horizontal axis) (*Cistus albidus*) (Ormeno et al., 2007). Top plot: SM effect on monoterpene emission fluxes of individual structured monoterpenes, as well as of the total sum of monoterpenes; lower plot: the same for individual structures and total sum of sesquiterpene emissions (*Rosmarinus officialis*) (Ormeno et al., 2007).**



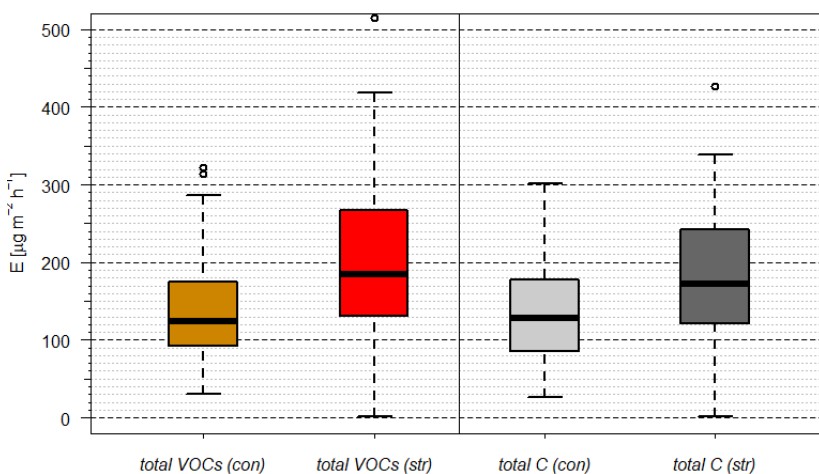

**Figure 5: Change for individual mono- and sesquiterpene species emission fluxes of *Fagus sylvatica* for two *SM* conditions**
655    **(Rombach, 2018). Because of not normal distributed data, values are stated as median and 25th (lower) and 75th (upper) percentiles.**



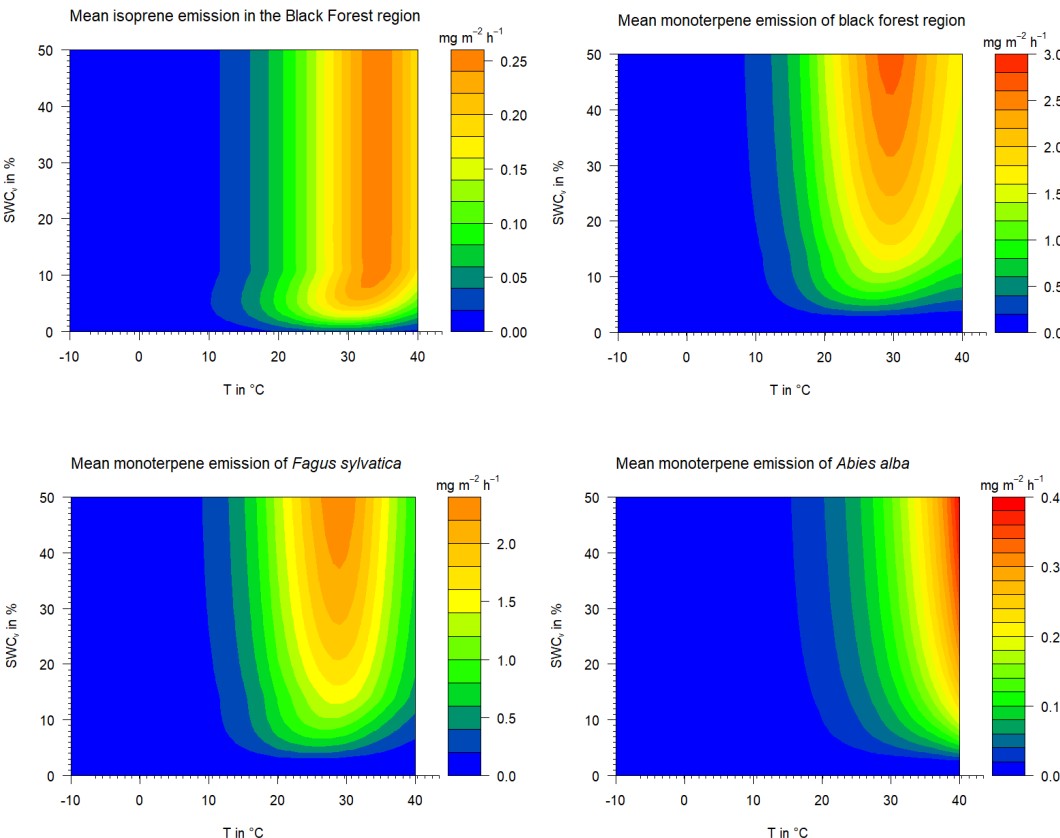

**Figure 6: Mean isoprene and monoterpene emissions under different temperature and drought stress conditions. "Black Forest region" represents the forest inventory mean tree species mixture of the Black Forest area. Upper left: total Isoprene emission, upper right: total monoterpene emission, baseline:** *Fagus sylvatica* **and** *Abies alba* **based MT emission rates (lower left and right).**


**Figure 7:** Calculated $\gamma_{SM}$ effects during 2016 at Freiamt on isoprene, total MT and SQT emission rates. Note, two induced drought periods during summer and quite saturated conditions else when.