# Peer review of "Biogenic isoprenoid emissions under drought stress: Different responses for isoprene and terpenes"

_Biogeosciences, 2019_

## Referee Comment (RC1) · Violeta Velikova (Referee) · 11 Jul 2019

The present manuscript addresses parameterization of isoprenoid emissions by vegetation under drought. The authors use available data on different tree types to describe the effect of soil water availability on biogenic emissions. Specifically, the present study is focused on the individual behaviour of isoprene, mono- and sesquiterpene exchange fluxes in correlation with soil moisture using different hypotheses: 1) a stepwise effect proposed by Guenther et al. 20006; 2) a growth rate bahaviour (in the present ms); 3) pattern of hydraulic conductivity (in the present ms); and 4) a stress defense response (in the present ms). The manuscript could contribute for better understanding

of processes controlling the biogenic emissions in conditions reflecting climate change. The authors correctly describe related studies and indicate their contribution. In this respect, the paper addresses relevant scientific question within the scope of Biogeosciences. The conclusions made are adequate and are supported by the data presented in the manuscript. The scientific methods are properly outlined. However, it is not clear how the watered (reference) Fagus sylvatica trees were separated from those to which water was not supplied at Freiburg nursery. For clarity, I would suggest more details about the location of the trees namely: 1) what is the distance between the two groups of trees? 2) How the water spreading into the soil from watered to non-watered side was prevented? 3) Taking into account the depth of F. sylvatica root system, is it enough to measure SWCv at the deepest of 75 cm in order to get correct effect of soil water availability on trees? The title reflects the paper content and the abstract provides adequate summary of the results. The manuscript is well structured. The figures are enough informative and with good quality. The used abbreviations are defined in Table 1. There is no need any parts of the paper to be changed. The quoted references are appropriate and reflect the "state of the art" in the field. The amount and quality of the supplementary material is adequate.

Minor comments: Line 47: Change Kelvin (K) to Celsius. Line 158-162: reference(s) has to be provided related to ROS detoxification/reduction of BVOCs. Line 203: after "(Eq. 5)" change to "c)" instead of "a)".

---

## Referee Comment (RC2) · Anonymous Referee #2 · 14 Jul 2019

This manuscript describes the drought response of BVOC, especially terpenoid, emissions with a focus on a German beech forest. BVOC emission drought response is an important topic that has received relatively little attention and is a subject suitable for readers of Biogeosciences. There are three main components of the paper: 1) a literature review of ∼13 BVOC drought studies, 2) the development and description of 3 terpenoid emission drought response parameterizations, and 3) observations at a site in southwestern Germany. Each of these components is a potentially worthwhile contribution to the paper but there are issues with each component that should be addressed before this paper is published.

[Figure]

General comments

Literature review: This is a significant effort given the lack of similar reviews on this topic. An especially valuable aspect is the compilation of literature data shown in figures 1,2 and 3. How were these data were obtained? i.e., were they from tables in the referenced papers, by digitizing figures in the papers, by contacting the authors of the referenced manuscripts? It would be especially useful contribution to the scientific community if the authors reported these data (all the values in figures 1, 2 and 3) in a table in the supplementary material so they could be used to compare with other studies. Almost all of the reviewed studies represent European sites/trees even though there are some BVOC emission drought response studies from other parts of the world (e.g., Asia, US, Israel, Australia). This may be reasonable given the focus on a site in Germany but this limitation should be mentioned in the text, and perhaps the title, to make it clear that this is not a comprehensive review but rather is focused on European vegetation. In addition, the authors should consider the different vegetation types in their analyses and indicate the different vegetation types (e.g., temperate, Mediterranean, etc) in the Figures etc.

Drought response equations: The authors propose three algorithms that they have labeled as "biological growth curve", "hydrological conductivity" curve and "oxidative stress" curves. The term "biological growth curve" is not a good fit here since that term refers to a change with time. Also, section 2.3 focuses on the different pathways for BVOC to escape from a plant (i.e., diffusion control or stomata control) but the authors do not demonstrate that this is the main control over the drought response of these emissions. The numerical form of these parameterizations are fine but the labels are misleading and the authors should discuss alternative controlling processes and provide more convincing evidence that the escape pathway is the important factor controlling drought response behavior.

Observations: Section 2.4.1 indicates that BVOC enclosure measurements were made on beech saplings and then mentions OH and ozone reactivity of observed emission

rates and corresponding forest air composition. But then there is no mention at all in the paper about forest air composition measurements and the only mention of the seedling BVOC enclosure measurements is in Figure S1 which is not discussed and does not appear to be relevant to the paper. These BVOC observations should either be removed from the paper or the presentation and discussion of this data should be substantially enhanced.

Specific comments:

line 16, "whereas of others": delete "of"

line 19: "On the contrary, OH and ozone reactivity enhance". Contrary to what?

line 26: "largest contribution to global carbon flux besides carbon dioxide and methane" seems to suggest that methane emission is larger than BVOC but the authors state an annual BVOC emission of >1 Pg which is higher than most (or all) methane emission estimates.

Line 137: What is meant by "least barrier"

Line 189: what is meant by " emissions of any vertical mixing "

Line 191: tenths or tens? What is the point of this sentence? Is it regarding characterization of the source footprint of the BVOC observed in the air? What is meant "next kilometers". Either better describe what is being said here or delete this sentence.

Line 230: "extend" => "extent"

Line 261, 331 Figure 2/3): It should be noted that Guenther 2006, 2012 (i.e., MEGAN2 and MEGAN2.1) applies this soil moisture algorithm ONLY to isoprene and did not recommend using it for monoterpenes or sesquiterpenes.

Line 293: ent-kaurene is not a sesquiterpene. It is a diterpene.

Line 310: fasted => faster

Line 344: What is the evidence for this generalization that overall BVOC increases with drought? An incorrect doi is listed for the "Rombach, J." reference (the one given is for the Lupke et al. 2017 paper).

Figure 5: The legend does not seem to describe what is shown in the figure. What are the two conditions? What are the individual fluxes?

Figure 6 and 7: Clarify whether these are model results or observations.

Figure 7: Show the induced drought periods.

Some sentences could be improved with editing for English usage. For example, sentences in line 20, 52, 79, 91, 92, 110, 149, 154, 176, 264, 284, 297, 322,339, 382, 410 and others

―――――――――――

---

## Author Comment (AC1) · 22 Aug 2019

General:

Thanks for the nice review and the questions raised. Since the details about the investigation conducted at the plant nursery in Freiburg in 2018 were rather scarce, we will add the following information to the present manuscript in line 170ff: "The distance between the edges of control and stressed group were approximately 10 m with 1 m distance between individual tree stems. Water was added by a watering pot by moderate flows to each of the trees in order to control the amount of water and to minimize the effects on neighboring trees.". The question concerning the soil water status of the

seedlings is important. We have made two approaches, i.e. a) measuring the pre-dawn water potential using a Scholander bomb as well as b) approximating the local soil water content by reference measurements oft he German Weather Service at the same soil conditions ca. 600~m in distance. The latter method used measurements down to 75 cm soil depth. We considered the water status of the seedlings as key parameter and transferred the derived values to soil water availability (SWA) taking the soil composition into account. By doing so, we hope to get close to the real SWA conditions and marked this by a notable errorbar, derived from the different approaches.

Regarding the minor comments:

L. 47: We'll replace "Kelvin" by "degree Celsius" as the units are only shifted by 273.15 K and slope is not affected.

L. 158-162: The following references will be added to "ROS detoxification/reduction of BVOCs": Niinemets et al. (2014), Parveen et al. (2018), Piechowiak et al. (2019) and Yalcinkaya et al. (2019).

L. 203: Will be done.

References:

Niinemets, Ü., Fares, S., Harley, P. and Jardine, K.J. (2014). Bidirectional exchange of biogenic volatiles with vegetation:emission sources, reactions, breakdown and deposition. Plant Cell Environ. 37, 1790–1809, doi: 10.1111/pce.12322. (already in the manuscript)

Parveen, S., Harun-Ur-Rashid, M., Inafuku, M., Iwasaki, H., and Oku, H. (2018). Molecular regulatory mechanism of isoprene emission under short-term drought stress in the tropical tree Ficus septica. Tree Phys. 39, 440-453, doi: 10.1093/treephys/tpy123. (new)

Piechowiak, T., and Balawejder, M. (2019). Impact of ozonation process on the level of selected oxidative stress markers in raspberries stored at room temperature. Food

Chem. 298, 125093, doi: 10.1016/j.foodchem.2019.125093 (new)

Yalcinkaya, T., Uzilday, B., Ozgur, R., Turkan, I., and Mano, J. (2019). Lipid peroxidation-derived reactive carbonyl species (RCS): Their interaction with ROS and cellular redox during environmental stresses. Environ. Exp. Bot. 165, 139-149, doi: 10.1016/j.envexpbot.2019.06.004. (new)

––––––––––––––––––––––––––––––––

---

## Author Comment (AC2) · 22 Aug 2019

First of all, thanks to the reviewer for the valuable comments made. We will address the individual issues step by step as listed in the review:

**Response to general comments:**

Indeed the studies used were digitalized by extracting the datasets from the individual publications as far as we could access it either personally or by web. It's a very good

idea to put all data into a datasheet for the community to use it henceforth. This will be done as suggested as an extra supplementary dataset.

It's correct that most of the datasets were obtained from Europe, as a lot of studies on drought effects were conducted especially in the Mediterranean and the project is focused on this area. However, looking for other studies with sufficient parameter measurements available for comparison were difficult to find. Potentially further information were obtained by the experimentalists but were not accessible within the article or supplementary datasets. An inclusion of further studies is welcome anytime. As suggested the different studies will be classified by area of investigation (Mediterranean, temperate) in the overview Table 2, where it is supposed to fit best. But one of the outcomes was actually that the plant response curves for individual species were physically i.e. molecular property based, but not species dependent. The plant species behavior was reflected in the selection of the different intensities of processes and may be adapted in this way to the local environmental conditions such as typical wetness, temperature and oxidation strength. However, we do not have sufficient data to clearly prove this hypothesis in a statistically significant way.

The term 'biological growth curve' is based on a variety of fit curves to match best with observations. The fitting equation was derived from 'biological growth' behavior, i.e. stresses to occur similar to the growth process. Those are supposed to be linked directly to processes involved in the plant status and establishing a metabolic balance. While some compounds such as isoprene may be less hindered by the stomata closure, its emission may be an appropriate adaption measure of the plant to different conditions (e.g. energy fluxes). We will consider your comments by adding an explanatory sentence at the introduction of the fitting curve term in L. 150.

Regarding section 2.3, i.e. the fitting curves and the named driving forces: We do

not have sufficient understanding of and information about the detailed physiological processes acting during production, emission and stress response in total. This is supposed to be gained in further experiments in the community. Thus, the present manuscript is only focused on figuring out the plant response to drought stress for different compound classes. A review covering all the named aspects would probably be very extensive and fill a book, which would be nice to have. The controlling factor of emissions can be added as follows: "Isoprene is known to be emitted close to production (Guenther et al., 1995; 2006; 2012) and is therefore controlled by the production process itself as well as by diffusion gradients between plant and atmosphere. The larger sized monoterpenes at least partially require passage through the stomata, as their size does not necessarily permit diffusion directly through the plants cuticula (Sharkey et al., 1991). Therefore monoterpene emissions are controlled by the stomata opening to a larger extend than isoprene and less by the production. Parts of the produced terpenes may be stored if not emitted for later usage. This is even more evident for sesquiterpenes with 50

Thanks for naming the issue of 'forest' air. This will be corrected to 'ambient' air, since the plant nursery is not placed in the forest but at the campus site. Figure S1 is added to the supplementary material, since it demonstrates a large variety of the behavior of different VOCs and no lumping of functional groups seems possible. It is not part of the paper, but to support following the analysis pathway. There is a tendency of an overall link predominantly by emission. Ozone and OH reactivity values were calculated based on reactivities as listed in Table S1 and the individual emission rates. Thus, no further measurements like nitrogen oxides etc was needed. Temperature measurements have been taken from the nearby German weather service station named in L. 172f and temperature will be added to this paragraph. The BVOC measurements have been described in L. 176ff and methods have been referred to. While a further discussion would cause the manuscript to increase notably the measurements are used here to demonstrate only the behaviour of the plants emission reactivity at declining water status in addition to the match with other observations with

respect to water availability and terpene emission rates.

**Response tp specific comments:**

L. 16: OK.

L. 19: Correct. "On the contrary..." will be changed to "On the contrary to declining soil water availability,..."

L. 26: Correct. To prevent any misunderstanding the term "and methane ($CH_4$)" will be deleted.

L. 137: It should have stated as "last" or "final barrier" before being released to the ambient. The 'e' will be deleted.

L. 189: Correct. "emissions of any vertical mixing" will be changed to "effects of any vertical mixing".

L. 191: Correct. It is tens not tenths and the second part of the sentence "...and the area of emission can be easily traced back within the next kilometers in distance (unpublished data)" will be deleted.

L. 230: Done.

L. 263/331. Thanks. This will be done to prevent misunderstanding and misusage.

L. 293: True. This with be included in L. 292 as "SQT and diterpene (DT) (....)".

L. 310: Thanks.

L. 344: We stated "tendency" as this was found in the Rombach study (Rombach, 2018) and aimed not to generalize in total. We will put the references next to tendency to indicate that this is based on a limited amount of data, to be investigated further.

The comment on the DOI is absolutely correct and the DOI will be removed from the thesis reference!

Fig. 5: The figure caption will be improved. Total monoterpene emission fluxes are plotted on the left, the corresponding of sesquiterpenes on the right. The individual emission fluxes will make the figure more complex but will be added in different colors. To display all 89 compounds identified would overload the plot. However we will include the dataset in the supplementary dataset to be uploaded for everyone interested in this study.

Figs. 6 and 7: These results are gained by applying the derived parameterisation for isoprene and monoterpene emission rates for a) European beech ($Fagus\ sylvatica$, lower plots) and for all species present in the Black Forest according to their basal emission rate. This will be clarified in the caption adding "...using the parameterisations of Guenther et al. (2012) and the SWA or SWC dependency derived in this study." In Fig. 6.

Fig. 7: Will be done by adding a shaded box for the time of enforced drought. We will ask for language editing.

Thanks a lot for the detailed suggestions made!

**References:**

Guenther, A., Hewitt, C. N., Erickson, D., Fall, R., Geron, C., Graedel, T., Harley, P., Klinger, L., Lerdau, M., McKay, W. A., Pierces, T., Scholes, B., Steinbrecher, R., Tallamraju, R., Taylor, J., and Zimmerman, P. (1995). A global model of natural volatile organic compound emissions. J. Geophys. Res., 100, 8873–8892, doi: 10.1029/94JD02950.

Guenther, A., Karl, T., Harley, P., Wiedinmyer, C., Palmer, P. I., and Geron, C. (2006). Estimates of global terrestrial isoprene emissions using MEGAN (Model of Emissions of Gases and Aerosols from Nature). Atmos. Chem. Phys., 6, 3181-3210, doi: 10.5194/acp-6-3181-505 2006.

Guenther, A. B., Jiang, X., Heald, C. L., Sakulyanontvittaya, T., Duhl, T., Emmons, L. K., and Wang, X.: The Model of Emissions of Gases and Aerosols from Nature Version 2.1 (MEGAN2.1) (2012). An extended and updated framework for modeling biogenic emissions. Geosci. Model Dev., 5, 1471–1492, doi; 10.5194/gmd-5-1471-2012.

Rombach, J. (2018). Einfluss reduzierter Wasserverfügbarkeit auf die VOC-Emissionen bei Buchen, Albert-Ludwigs Universität, Freiburg, 13, 14, doi: 10.1186/s13007-017-0166-6.

Sharkey, T.D., Holland, E.A., and Mooney, H.A. (1991). Trace gas emissions by plants. Academic Press, London, UK.